# TSD-Truncated Structurally Aware Distance for Small Pest Object Detection

**DOI:** 10.3390/s22228691

**Published:** 2022-11-10

**Authors:** Xiaowen Huang, Jun Dong, Zhijia Zhu, Dong Ma, Fan Ma, Luhong Lang

**Affiliations:** 1Hefei Institutes of Physical Science, Chinese Academy of Sciences, Hefei 230031, China; 2University of Science and Technology of China, Hefei 230026, China; 3Anhui Zhongke Deji Intelligence Technology Co., Ltd., Hefei 230045, China; 4Wuhu Institute of Technology, Wuhu 241006, China

**Keywords:** small object detection, pest detection, truncated structurally aware distance, truncated structurally aware loss, faster R-CNN

## Abstract

As deep learning has been successfully applied in various domains, it has recently received considerable research attention for decades, making it possible to efficiently and intelligently detect crop pests. Nevertheless, the detection of pest objects is still challenging due to the lack of discriminative features and pests’ aggregation behavior. Recently, intersection over union (IoU)-based object detection has attracted much attention and become the most widely used metric. However, it is sensitive to small-object localization bias; furthermore, IoU-based loss only works when ground truths and predicted bounding boxes are intersected, and it lacks an awareness of different geometrical structures. Therefore, we propose a simple and effective metric and a loss function based on this new metric, truncated structurally aware distance (TSD). Firstly, the distance between two bounding boxes is defined as the standardized Chebyshev distance. We also propose a new regression loss function, truncated structurally aware distance loss, which consider the different geometrical structure relationships between two bounding boxes and whose truncated function is designed to impose different penalties. To further test the effectiveness of our method, we apply it on the Pest24 small-object pest dataset, and the results show that the mAP is 5.0% higher than other detection methods.

## 1. Introduction

Object detection is one of the key issues in computer vision tasks, and as the cornerstone of image understanding and computer vision, object detection is the basis for solving higher-level vision tasks such as segmentation, scene understanding, object tracking, image description and event detection. Object detection has also received considerable research attention for decades. Even though massive research methods have achieved significant progress, most of them are devoted to detecting objects of normal size. Small objects (less than $32^{2}$ pixels) usually struggle to show sufficient appearance information, which increases the difficulty of feature recognition and extraction, leading to poor performance when detecting small objects. However, small objects extensively exist in the real world and play a crucial role in many fields, such as automatic driving [1,2,3], defect detection [4,5,6], remote sensing [7,8,9] and pest detection [10,11,12]. In particular, agricultural production is greatly restricted by the erosion of crop pests, which has brought infinite losses to the global agricultural economy. Therefore, introducing deep learning and pest detection to smart agriculture is essential.

Existing detectors have achieved significant progress when detecting objects of normal size; however, their application in small objects has not done the same, and pest detection thus still faces massive challenges:Few available features. Due to the small size of objects, in the process of feature extraction, as the number of CNN layers increases, the target feature information tends to be weakened layer by layer, making it difficult to extract discriminative features. Further, in the context of a multi-layer network, it may also cause the missed detection of some objects.High positioning accuracy requirements. Because the small objects occupy a small area in the image, it is more difficult to locate their bounding box than for objects of normal size. In addition, in the anchor-based detector, the number of anchors matching small objects during training is much lower than that of usual-scale objects, which also makes small objects more difficult to detect to some extent.Small objects are gathering. Firstly, due to the habits of pests, it is easy for them to gather together under the catching device. Secondly, after multiple convolutions, the adjacent small objects in the aggregation area will be aggregated into one point on the feature map of the later layer, which will make the detection model unable to distinguish objects effectively. Lastly, if the distance between the small objects in the aggregation area is too close, this will make it difficult for the bounding box to regress, and the model will struggle to converge.

Consequently, recent research for small object detection has mainly concentrated on improving feature discrimination. Constructing feature-level pyramids of different scales has become an excellent method in recent years. For example, the YOLO series [13,14,15,16] took an end-to-end method to optimizing detection performance. Single-shot MultiBox detector (SSD [17]) can detect objects in feature maps of different resolutions. Spatial pyramid pooling (SPP [18]), feature pyramid network (FPN [19]), BiFPN [20], and recursive-FPN [21] construct a top-down structure with lateral connections to combine different scales of feature information to improve object detection performance. Multi-scale feature learning has also been widely used in pest detection. Ref. [22] proposed a multi-scale feature extraction module by using multi-layer dilated convolution. This method effectively reduced the training time. Ref. [23] proposed a multi-scale super-resolution (MSR) feature enhancement module; this method enhanced the feature expression ability using a feature fusion mechanism. However, this super-resolution feature enhancement module also incurs extremely high computational costs and more training time.

Perceptual generative adversarial networks(Perceptual GAN [24]) were the first attempt to apply GANs to small object detection, which improved small object detection by reducing the representation difference between small objects and large objects. In agriculture, there are not enough public datasets, which reduces the data-fitting ability of detectors to some extent. Therefore, Ref. [25] utilized a conditional GAN to generate synthetic images of tomato plant leaves and then trained a detector on synthetic and real images. Ref. [26] proposed unsupervised activation reconstruction GAN to learn one-to-one mapping for tomato plant disease images by aggregating cycle consistency and activation reconstruction loss. However, the application of GAN in agriculture usually focuses on tomato images and tomato disease images due to their distinctive features. A visible flaw is that images generated by a GAN lack structural definition for an object of interest, which indicates that GANs are not suitable for pest detection owing to the inconspicuous features presented by pests in the images.

Indeed, some researchers have proposed abandoning the mechanism of anchors directly and have converted the small-object detection task into the estimation of key points [27,28]. This method also has been introduced into agriculture. Ref [29] designed an anchor-free network to predict the position of a dairy goat in the search frame; it effectively circumvents the shortcomings of the fixed anchor ratio, but it resulted in a high-recall but low-precision test result.

Despite these different detectors, evaluation metrics for bounding box regression plays a crucial role in predicting a bounding box. In terms of evaluation metrics, IoU is the most widely used method for label assignment, non-maximum suppression (NMS) and bounding box regression loss function. However, IoU does not work well when an object’s size is too small because of its sensitivity to small objects’ short location deviation. Many methods have been proposed by adding some penalty terms based on IoU [30,31]; their graphical descriptions are as shown in Figure 1. However, a penalty term has a limited impact on the metric value, and the IoU still plays a decisive role. Therefore, these methods also have the same disadvantages of the IoU metric.

The normalized Wasserstein distance (NWD [32]) was proposed to adapt to the characteristics of small and tiny objects. It firstly uses a 2D Gaussian distribution to model the bounding boxes and then proposes a new measurement method called NWD to calculate the similarity between them by their corresponding Gaussian distributions. However, during the modeling of the bounding boxes as 2D Gaussian distribution processes, some of the pixels in the bounding box are discarded to some extent. For small objects with few pixels, discarding some pixels may be decisive. In addition to this, the computational cost of the second-order Wasserstein distance between two 2D Gaussian distributions was excessively high.

Observing that IoU and NWD did not perform well in pest detection, we propose a new metric to adapt to the characteristics of small pests. The contributions of this paper are summarized as follows:A new metric, truncated structurally aware distance (TSD), is proposed to measure the similarity of bounding boxes and replace IoU in the label assignment. TSD simply uses the standardized Chebyshev distance to describe the similarity of bounding boxes (as shown in Figure 1), which could solve the problem regarding IoU sensitivity to small objects’ localization bias.Instead of using loss=1−TSD, we design a new loss function dubbed TSD loss based on TSD; it can use the truncated method to describe the structural regression loss for small pests.The proposed TSD can significantly improve the performance of the network for small object detection in popular anchor-based detectors, and the performance is improved from 46.0% to 50.7% on Faster R-CNN on the Pest24 dataset.

This article is composed of five sections. Section 1 introduces the difficulties and necessities in pest detection, and presents the related work, Section 2 mainly introduces the new method proposed in this paper for small-object pest detection, Section 3 lists the relevant experimental results, and Section 4 summarizes some of the advantages and disadvantages of this work.

## 2. Methodology

To adapt to the characteristics of small-object detection, a new method must be designed to improve the detection accuracy on Pest24 to meet the needs of intelligent detection of pest targets.

### 2.1. Truncated Structurally Aware Distance

IoU was the most widely used metric for label assignment:(1)$$IoU=\frac{|B\cap B^{gt}|}{|B\cup B^{gt}|}$$
where $B^{gt}=\left(x^{gt},y^{gt},w^{gt},h^{gt}\right)$ is the ground truth, and B=(x,y,w,h) is the predicted box. As shown in Figure 2, IoU only works when the bounding boxes are intersected. The values are always 0 when they are separated, and the values are equal when they include each other.

Therefore, we firstly propose a new metric called TSD to replace the IoU metric. It is defined as:(2)$$TSD=1-\sqrt{\frac{D_{Chess}^{2}\left(b,b^{gt}\right)}{S}}$$
where $D_{Chess}\left(b,b^{gt}\right)=max(|x_{1}-x_{2}\left|,\right|y_{1}-y_{2}\left|\right)$ represents the Chebyshev distance between the center points $\left(x_{1},y_{1}\right)$ and $\left(x_{2},y_{2}\right)$ of two bounding boxes, and *S* represents the average area of all ground truths in the dataset. Therefore, $\sqrt{S}$ can be expressed as the average side length of all ground truths, and consequently, the Chebyshev distance can be standardized to describe the distance of two bounding boxes and has a canonical form. Its value range is (−∞,1]. When the two center points of two bounding boxes are coincident, the distance is 1. When the two center points of two bounding boxes are far apart, the TSD tends to be infinitely small. TSD only focuses on the standardized Chebyshev distance of two center points, and it represents the ratio of the distance between two center points to the average side length to some extent. It shows that this method is more suitable for small objects whose total size is less than 32 pixels.

As shown in Figure 3, without loss of generality, we will discuss the changes in the indicator values in the following two cases.

We set the side length of B to be half of the side length of A, and draw B away from A along the diagonal lines of A. Respectively, we can observe that the IoU is too sensitive to the position bias of small objects. Compared with the IoU, it can be seen that the four curves of TSD are completely coincident, which indicates that TSD is not sensitive to the size of the bounding boxes.

Firstly, the metric’s change in bounding boxes with different sizes under the same positional bias suggests that the distinction between positive/negative samples under TSD may be better than IoU. Furthermore, as shown in Figure 2, the values of TSD are different when two bounding boxes are separated, whichis related to the distance of center points: the farther they are, the smaller the value. When the bounding boxes contain each other and the center points are completely coincident, the value is 1, but it is not 1 in other cases. Accordingly, The features selected by the bounding box are gradually reduced from the inside to the outside. Therefore, TSD has a superior performance to IoU in bounding box matching and selection.

### 2.2. Truncated Structurally Aware Distance Loss

The smooth L1 loss function [33] was regularly used in Faster R-CNN when calculating the bounding box regression loss:(3)$$Smooth_{l1}=\left\{\begin{array}{cc} 0.5x^{2}, & \;|x|<1,\\ |x|-0.5, & \;otherwise. \end{array}\right.$$

In this function, the two points $\left(x_{1},y_{1}\right)$ and $\left(x_{2},y_{2}\right)$ of the bounding boxes are regarded as independent points, which does not consider the correlation between the vertices of the bounding box. Moreover, this function cannot be consistent with the evaluation index used by the bounding box matching, resulting in an inaccurate regression loss.

Combined with the proposed TSD, a matching regression loss function that was called truncated structurally aware distance loss (TSD loss) was designed in this paper. When describing the TSD loss, it is defined as:(4)$$TSD_{loss}=\left\{\begin{array}{cc} \sqrt{\frac{D_{Chess}^{2}\left(b,b^{gt}\right)}{S}}, & \;separated,\\ 1-\cos\theta , & \;intersect,\\ 1-e^{-|r_{1}-r_{2}|}, & \;contain. \end{array}\right.$$
where $D_{Chess}\left(b,b^{gt}\right)=max(|x_{1}-x_{2}\left|,\right|y_{1}-y_{2}\left|\right)$ represents the Chebyshev distance between the center points of two bounding boxes $\left(x_{1},y_{1}\right)$ and $\left(x_{2},y_{2}\right)$, and *S* represents the average area of all ground truths in the dataset. $r_{1}$ and $r_{2}$ represent the radius of the circumcircle of the two bounding boxes. According to the cosine theorem:(5)$$\cos\theta =\frac{r_{1}^{2}+r_{2}^{2}-d^{2}}{2*r_{1}*r_{2}}$$
where *d* represent the distance between the center points of the two bounding boxes. The graphical description of each segment of the loss function is shown in Figure 4.

As shown in Figure 4a, the distance between the two center points plays an important role in the calculation of regression loss. The shorter the distance, the smaller the distinction between the two bounding boxes. As shown in Figure 4b, cosine similarity is used to describe the coincidence between the two bounding boxes. The smaller the θ, the higher the degree of coincidence between the two bounding boxes. As shown in Figure 4c, the distinction between $r_{1}$ and $r_{2}$ can be seen as the degree of fitting of the two bounding boxes.

*X* is set as the independent variable of each truncated function in Equation (Equation 4), and another form of Equation (Equation 4) can be obtained:(6)$$TSD_{loss}=\left\{\begin{array}{c} \sqrt{\frac{D_{Chess}^{2}\left(b,b^{gt}\right)}{S}}\\ 1-cos\theta \\ 1-e^{-|r_{1}-r_{2}|} \end{array}\right.=\left\{\begin{array}{cc} x, & \;separated,\\ 1-cosx, & \;intersect,\\ 1-e^{-x}, & \;contain. \end{array}\right.$$
where Equation (Equation 7) is derived from x to obtain:(7)$$\frac{dTSD_{loss}}{dx}=\left\{\begin{array}{cc} 1, & \;separated,\\ sinx, & \;intersect,\\ e^{-x}, & \;contain. \end{array}\right.$$

When two bounding boxes are intersected or contain each other, the gradient to *x* will be small, and their value range is [−1,1] and (0,1]. When two bounding boxes are separated, the upper limit of the gradient to *x* is 1, which is not so large as to destroy the network parameters.

Position deviations of small objects require more accurate loss calculations. Therefore, according to different structural relationships, using truncated structurally aware functions to calculate regression loss is more suitable for small objects.

### 2.3. Detector

Small objects have fewer available pixels, making it challenging to extract useful features from them. As the number of network layers increase, the features and location information of small objects are gradually lost. The above characteristics all lead to both deep semantic information and shallow representation information, which are required when detecting small objects. We observed that multi-scale learning combines these two, which is an effective strategy to improve the performance of small pest object detection.

Feature pyramid network (FPN [19]) is a well-designed multi-scale detection method, and it is a general architecture that can be used in conjunction with various backbone nets. In our paper, ResNet [34] and FPN [19] have been combined to generate feature maps as our backbone. Our ResNet-FPN structure includes three parts: a bottom-up connection, a top-down connection and a horizontal connection (as shown in Figure 5).

Bottom-up. Specifically, according to the size of the feature map, ResNet is divided into four stages as the backbone network: Stage2, Stage3, Stage4, and Stage5. Each stage outputs Conv2, Conv3, Conv4 and Conv5 in its last layer, and these output layers are defined as $C_{2}$, $C_{3}$, $C_{4}$, $C_{5}$. It is a simple feature extraction process.Top-bottom. Up-sampling starts from the highest layer. The nearest neighbor up-sampling method instead of a deconvolution operation has been used directly in our up-sampling process.Horizontal connection. The up-sampled results are fused with the feature map generated from the bottom-up process. After the fusion, the fused features are processed by the convolution kernel to eliminate the aliasing effect of up-sampling.

Afterwards, $\left\{P_{2},P_{3},P_{4},P_{5},P_{6}\right\}$ become the input of RPN, and the inputs of R-CNN are $\left\{P_{2},P_{3},P_{4},P_{5}\right\}$. This structure can merge the characteristics of each level, so that it has semantic solid information and spatial solid information at the same time.

As the masterpiece of two-stage object detection algorithms, Faster R-CNN has achieved many examples of excellent performance in object detection. In our paper, we firstly input the images into the backbone network to generate feature maps of the images. Secondly, applying the Region Proposal Network (RPN) [35] on the feature map, the proposed TSD is used to compute the similarity of predicted bounding boxes and the ground truths, and the proposed TSD loss is used to compute the regression loss (as shown in Figure 5). RPN can return rough proposals and corresponding scores. Thirdly, the Rol Align layer is used to modify all proposals to the same size. Finally, the proposals are passed to the fully connected layer to classify the object and predict the bounding box. The global network architecture is shown in Figure 5.

## 3. Experiment

### 3.1. Dataset

In this paper, the Pest24 dataset was recruited [36]. All the data from the Pest24 dataset were collected by the professional automatic pest image acquisition equipment developed by the Institute of Intelligent Machines, Chinese Academy of Sciences. Pest24 contains 25,378 multi-pest images with a resolution of 2095 × 1944 pixels. It includes 24 categories, including ultra-small object sizes, with a dense distribution of objects in each image, a high similarity of pest objects to each other in shape and color, and many object adhesions in pictures.

We used the same evaluation metrics as the MS COCO dataset [37], including $AP_{0.5:0.95}$, $AP_{0.5}$, $AP_{0.75}$, $AP_{s}$, $AP_{m}$, and $AP_{l}$. Specifically, $AP_{0.5:0.95}$ is the different IoU threshold, IoU = 0.5, 0.55,…, 0.95; average mAP, $AP_{0.5}$ and $AP_{0.75}$ are APs with IoU thresholds of 0.5 and 0.75. $AP_{s}$ is the average precision for small objects ($area\leq 32^{2}$), $AP_{m}$ is the average precision for medium objects ($32^{2}<area\leq 96^{2}$), and $AP_{l}$ is the average precision for large objects ($area>96^{2}$).

In addition, as shown in Figure 6, there is still a lack of labeled dataset instances. Only about 40% of the instances contain labels in an image, and the omission rate is too high, resulting in a low MAP value.

### 3.2. Implementation Details

We conducted all the experiments on a computer with a 1 NVIDIA GeForce RTX 3080 GPU, and Faster-RCNN was trained with stochastic gradient descent (SGD). We used batchsize=8, epoch=21, an initial learning rate of 0.01, and iterations every 3 epochs, each time adjusting to 0.33 times the initial learning rate. The weight decay was 0.0001, and the momentum was 0.9. The default anchorsizes=((8,),(16,),(32,),(64),(128)), aspectratios=(0.5,1.0,2.0). The number of proposals generated by RPN was 2000. Horizontal image flipping was the only form of data augmentation. Training loss was the sum of the classification loss and regression loss.

### 3.3. Ablation Study

***Focal loss.*** Using the focal loss [38] as the loss for the output of the classification sub-net, the total focal loss of the image was calculated as the sum of the focal losses on all anchors, normalized by the number of anchors assigned the ground-truth box. Since most anchors are easily negative, and the loss obtained under focal loss is negligible, we normalized using the number of specified anchors instead of the total anchors. We note that the weight α assigned to the rare class has a stable range, which interacts with γ, so these two must be chosen together (as in Table 1), and finally, the best results are obtained when α=0.5,γ=2. This indicates that the number of samples in different categories on Pest24 dataset is balanced to some extent.

***Different parts of the detector.*** Faster R-CNN consists of two networks: RPN and R-CNN. Both of them include label assignment, NMS and a regression loss function. TSD is applied in the RPN module, and then rough proposals are generated and binary labels are assigned to the anchors for training the classification and regression heads. We gradually replaced IoU with TSD in these modules. As shown in Table 2, the comprehensive performance of detectors gradually improves when applying TSD in more parts. Additionally, the AP value reaches its highest value when applying TSD in all three modules. It can also be observed that $AP_{s}$ decrease when applying TSD in the NMS module. This indicates that the application of standardized Chebyshev distance in NMS ignores slight differences to some extent and needs further improvements.

***AP for each category.*** The Pest24 dataset has 24 different categories. The $AP_{0.5}$ values of each category after applying TSD to the label assignment, NMS and regression loss modules on the test set are as shown in Figure 7. It can be seen that the AP values are well balanced across categories of different sizes, which shows that the proposed method is appropriate for objects of various scales.

### 3.4. Comparison of Different Metrics

The similarity between bounding boxes is usually measured by some IoU-based metrics. In this paper, when Faster R-CNN was used to detect pest objects, we re-implemented the above four indicators (i.e., IoU, Generalized IoU (GIoU), Complete IoU (CIoU) and Distance-IoU (DIoU)) for the label assignment, NMS and regression loss functions, and the proposed TSD for comparison, specifically, the experimental results on the Pest24 dataset, are shown in Table 3. This table shows that the best performance is achieved by the proposed methods, reaching values of 51.0. We can also observe that imposing different penalties based on IoU did not significantly affect the results.

### 3.5. Comparison of Different Detectors

To verify the effectiveness of the method in this paper on pest detection, we selected some baseline detectors to detect the Pest24 dataset, respectively. Experimental results are shown in Table 4. It can be seen that none of the current state-of-the-art detectors have high mAP, which indicates that they do not perform well in the detection of small pest objects. We can observe that NWD attains the best performance on $AP_{s}$ but it did not do well in other metrics, which shows that it does not perform well in multi-scale detection. As a result, the proposed Faster R-CNN based on TSD and TSD loss achieves better results than SSD [17], RetinaNet [38], YOLO [14], Faster R-CNN [39], Cascade R-CNN [40], NWD [32] and DotD [41], which was proposed to detect tiny objects.

To explain the improvement of the proposed method in more detail, the curves of training loss and $AP_{0.5}$ of different detectors are shown in Figure 8. It is clear that compared with other detectors, the proposed TSD has a more excellent data fitting ability and is capable of small pest detection. In addition, the proposed TSD’s convergence rate is the fastest.

To visually observe the results, we visualize the detection results of the IoU-based Faster R-CNN (the second row) and the TSD-based Faster R-CNN (the third row) on the Pest24 datasets, as shown in Figure 9. Among them, the density of the object decreases column by column (the first column shows the densest distribution of pest images, and the fourth column show the sparsest distribution of pest images). Compared with the ground-truth (the first row), the visualization shows that the IoU-based detector exhibits missed bounding boxes (especially in dense pest images, as shown in the first column in Figure 9). We can also observe that the proposed TSD-based method can significantly improve true positives (TP) compared to the IoU-based method.

## 4. Conclusions

In this paper, a new metric and a new regression loss function are proposed. This new metric, TSD, simply uses standardized Chebyshev distance to describe the similarity of bounding boxes, which can replace IoU in label assignment and overcome the shortcomings of IoU sensitivity to small-object localization bias. The proposed new regression loss, TSD loss, takes the different geometrically structured relationships between two bounding boxes into full consideration and designs a truncated function to impose different penalties under different structures. The model that deployed the proposed method in the RPN module was trained on the Pest24 dataset. The experimental results show that the precision reached 51.0%, which is 5.0% higher than the model that deployed IoU. Moreover, the precision is higher than other widely used detectors. Additionally, the proposed methods are plug-and-play, and thus, it is convenient to apply the proposed method in practice.

Faster R-CNN is a two-stage object detector. RPN generates rough region proposals in the first stage, and R-CNN generates more refined classification and regression results in the second stage. More experimental results show that TSD is more suitable for rough region proposal generation applied in the first stage, and can maximize true-positive proposals (TP). How to design a new metric that can be applied in the second stage is also our future focus.

## Figures and Tables

**Figure 1 sensors-22-08691-f001:**
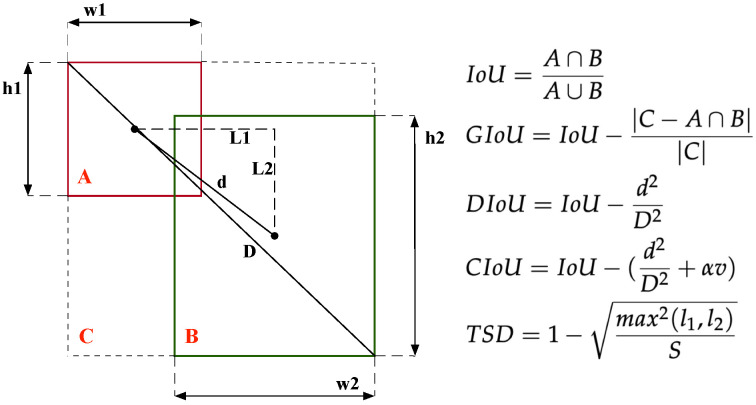
The graphical description of different metrics. Box A denotes the ground truth; box B denotes the predicted bounding box. $v=\frac{4}{\pi^{2}}{\left(arctan\frac{w_{A}}{h_{A}}-arctan\frac{w_{B}}{h_{B}}\right)}^{2}$, $\alpha =\frac{v}{(1-IoU)+v}$.

**Figure 2 sensors-22-08691-f002:**
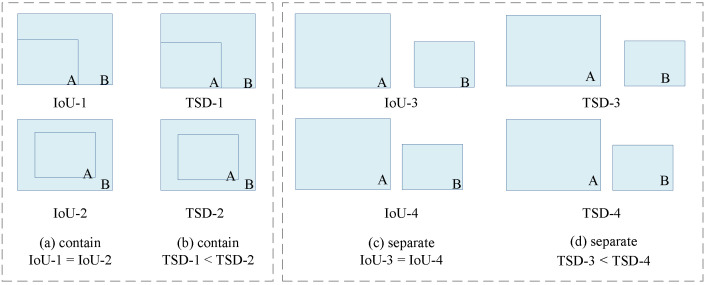
A comparison between IoU metrics and TSD metrics on different positional relationships. (**a**,**b**) represent the containment relationship, and (**c**,**d**) represent the separation relationship.

**Figure 3 sensors-22-08691-f003:**
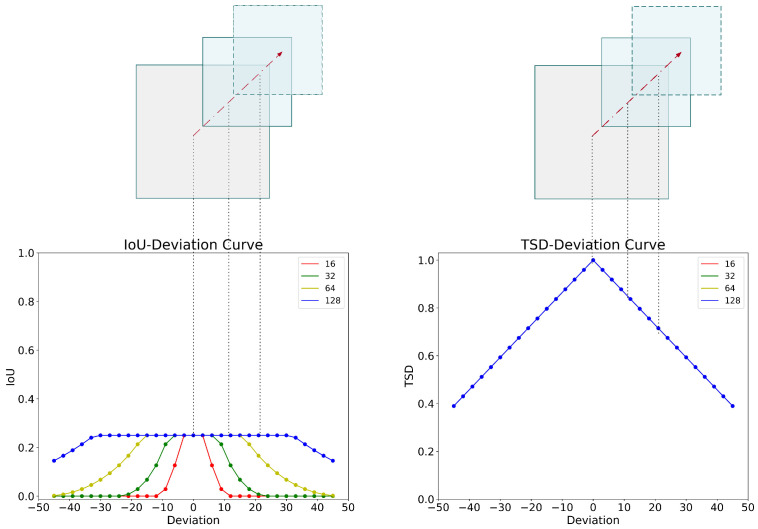
A comparison between IoU-deviation curve and TSD-deviation curve in four different scenarios. The abscissa value denotes the number of pixels of deviation between the center points of A and B, and the ordinate value denotes the corresponding metric value. On the left is the IoU-deviation curve, and on the right is the TSD-deviation curve.

**Figure 4 sensors-22-08691-f004:**
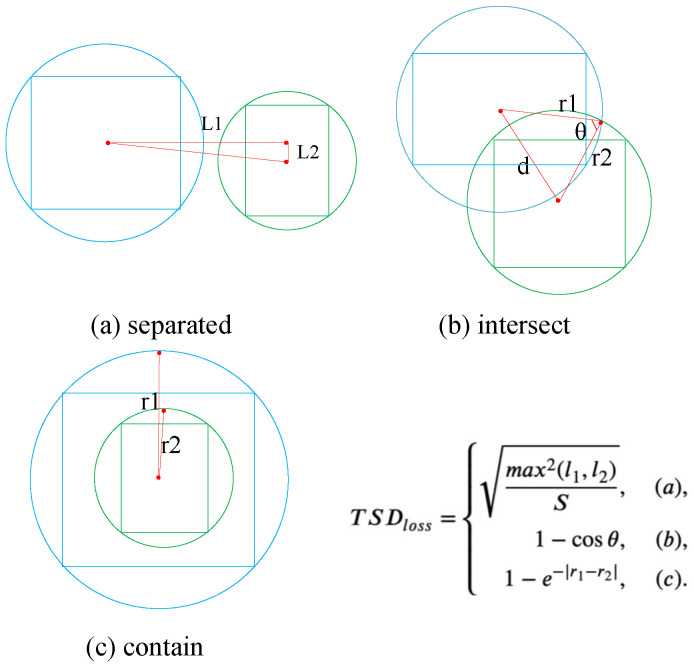
The graphical description of each stage loss in the TSD Loss function.

**Figure 5 sensors-22-08691-f005:**
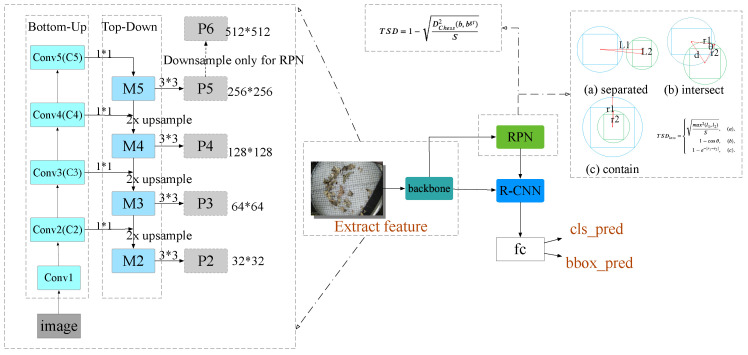
Network architecture. We use ResNet-50-FPN as the backbone of the two-stage network; the proposed TSD and $TSD_{loss}$ has been used in the RPN.

**Figure 6 sensors-22-08691-f006:**
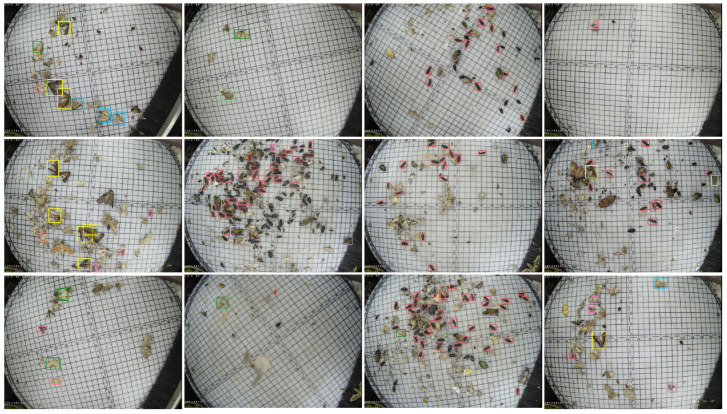
The samples of some ground truth boxes were labeled in the dataset, but not all the instances of the data are labeled in the dataset. There are many omissions in each image, resulting in a low MAP value.

**Figure 7 sensors-22-08691-f007:**
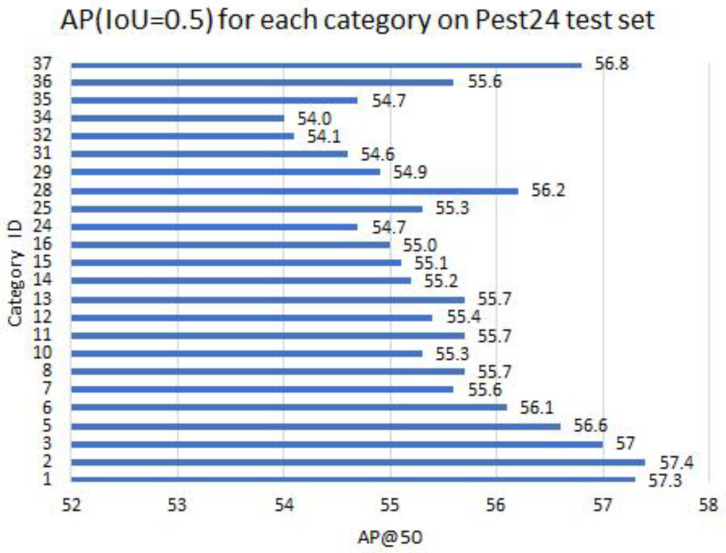
AP (IoU = 0.5) for each category on Pest24 test set.

**Figure 8 sensors-22-08691-f008:**
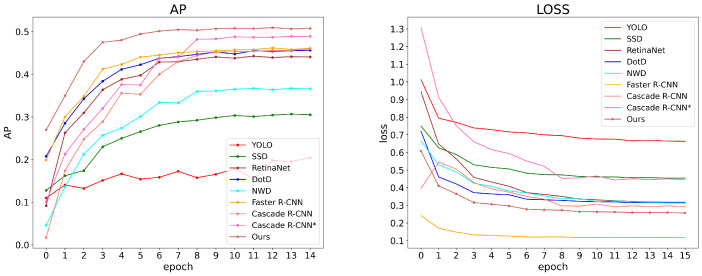
The AP curve (**left**) and training loss curve(**right**) on Pest24 dataset; they represent the results of different detectors (the proposed method is marked with *).

**Figure 9 sensors-22-08691-f009:**
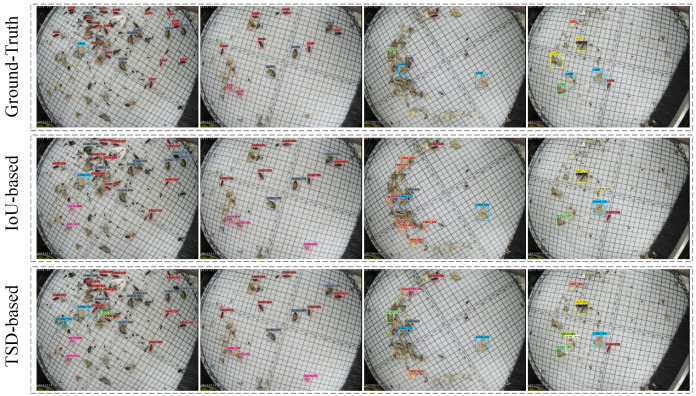
The visualization results based on IoU metrics and TSD metrics. The first row represents the ground truth of the dataset, and the second row represents the results under IoU, while the third row represents the results under the proposed approach.

**Table 1 sensors-22-08691-t001:** Comparison of different γ/α settings on Pest24 validation set.

Method	α	γ	${\mathrm{AP}}_{0.5}$	${\mathrm{AP}}_{0.5:0.95}$	${\mathrm{AP}}_{0.75}$	${\mathrm{AP}}_{s}$	${\mathrm{AP}}_{m}$	${\mathrm{AP}}_{l}$
CE Loss			48.2	29.5	33.3	17.2	35.7	70.0
Focal Loss	0.25	1	37.9	23.3	26.4	13.1	30.2	60.0
	0.25	2	37.8	23.2	26.2	13.0	29.8	70.0
	0.25	3	36.9	22.7	25.7	13.0	29.4	70.0
	0.5	2	**51.0**	**31.0**	**36.0**	**17.3**	**37.6**	70.0
	0.5	3	46.7	28.4	32.3	16.1	35.0	**80.0**

**Table 2 sensors-22-08691-t002:** Ablation study on Pest24 validation set.

Detector	Assigning	NMS	Loss	${\mathrm{AP}}_{0.5}$	${\mathrm{AP}}_{0.5:0.95}$	${\mathrm{AP}}_{0.75}$	${\mathrm{AP}}_{s}$	${\mathrm{AP}}_{m}$	${\mathrm{AP}}_{l}$
Faster R-CNN	TSD	IoU	$SmoothL_{1}$	50.3	30.3	33.4	18.2	36.3	70.0
	TSD	TSD	$SmoothL_{1}$	50.9	30.8	34.6	17.8	36.9	70.0
	TSD	IoU	TSD	50.7	30.7	34.7	**19.3**	37.2	60.0
	TSD	TSD	TSD	**51.0**	**31.0**	**36.0**	17.3	**37.6**	**70.0**

**Table 3 sensors-22-08691-t003:** Comparison of different metrics on Pest24 validation set.

Metrics	mAP					
	${\mathrm{AP}}_{0.5}$	${\mathrm{AP}}_{0.5:0.95}$	${\mathrm{AP}}_{0.75}$	${\mathrm{AP}}_{s}$	${\mathrm{AP}}_{m}$	${\mathrm{AP}}_{l}$
IoU	46.0	26.9	28.9	16.3	32.2	50.0
GIoU	43.8	23.4	22.9	14.0	27.5	40.0
DIoU	45.4	24.5	23.7	15.1	29.2	40.0
CIoU	46.1	24.9	24.0	16.2	29.3	30.0
Ours	**51.0**	**31.0**	**36.0**	**17.3**	**37.6**	**70.0**

**Table 4 sensors-22-08691-t004:** Quantitative comparison of the baselines and the proposed method (with *) on Pest24 validation set.

Method	Backbone	${\mathrm{AP}}_{0.5}$	${\mathrm{AP}}_{0.5:0.95}$	${\mathrm{AP}}_{0.75}$	${\mathrm{AP}}_{s}$	${\mathrm{AP}}_{m}$	${\mathrm{AP}}_{l}$
SSD [17]	ResNet-50	30.7	17.1	17.5	7.0	22.8	50.0
RetinaNet [38]	ResNet-50-FPN	44.3	27.0	30.1	15.1	33.1	50.0
YOLO [14]	DarkNet53	22.4	11.5	9.8	6.8	15.2	40.0
DotD [41]	ResNet-50-FPN	45.5	26.8	28.8	16.1	32.1	60.0
NWD [32]	ResNet-50-FPN	36.7	17.5	13.6	**31.8**	37.2	10.0
Cascade R-CNN [40]	ResNet-50-FPN	46.0	25.4	25.1	15.0	31.0	50.0
Cascade R-CNN *****	ResNet-50-FPN	48.9	27.7	28.3	15.2	34.2	60.0
Faster R-CNN [39]	ResNet-50-FPN	46.0	26.9	28.9	16.3	32.2	50.0
	MobileNetV2	41.8	22.5	22.3	11.5	29.2	50.0
	VGG16	47.1	27.3	29.2	16.8	33.3	40.0
Faster R-CNN *****	ResNet-50-FPN	**51.0**	**31.0**	**36.0**	17.3	**37.6**	**70.0**

## Data Availability

The code is available at https://github.com/baidawu/TSD-TSD-for-small-pest-object-detection, and accessed on 16 October 2021.

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
