# Peer review of "TSD-Truncated Structurally Aware Distance for Small Pest Object Detection"

_sensors, 2022, doi:10.3390/s22228691_

Round 1

Reviewer 1 Report

The article is about the detection of pest objects. A simple and effective metric and a loss function based on this new metric, Truncated Structural Aware Distance (TSD) was proposed.

The strength of the article is the numerical assessment of pest detection. This method was compared with other known methods of this type.

The article will be better if authors add more information:1. Please add information about the measurement conditions (temperature, humidity).

The test is carried out on pests living in various weather conditions. Does temperature and humidity affect the effectiveness of the research? Does the percentage effectiveness of the method change?

2. Please add some references. Some readers may not understand the essence of the research. Please add more references to help you understand the issue.

If the changes are introduced, I can accept the article.

Author Response

Dear reviewer,

Thank you for giving us the opportunity to submit a revised draft of the manuscript "TSD-Truncated Structural Aware Distance for Small Pest Object Detection". We appreciate the time and effort that you dedicated to providing feedback on our manuscript and are grateful for the insightful comments and valuable improvements to our paper. We have incorporated most of the suggestions made by the reviewer. Those changes are highlighted within the manuscript. Please see the attachment, for a point-by-point response to your comments and concerns. All page numbers refer to the revised manuscript file with tracked changes.

Reviewer 2 Report

Many excellent algorithms have appeared in the field of target detection and have been successfully applied in the industry. However, the problem of poor target detection rate of small targets has not been solved. This study serves the target detection of small pest, and proposes a new metric Truncated Structural Aware Distance (TSD). The loss function is designed to train Faster R-CNN. We can be sure that this is a meaningful work. This study discussed different metrics and different detectors respectively, and finally concluded that TSD based Faster R-CNN is effective in small pest target detection.

However, we also have some questions about some details of the article:

1. In the formula of TSD (formula 2), why use S (the average area of all ground truths in the dataset) as the denominator? Is the S value of the same dataset uniform? What is the purpose of this design? Please explain.

2. About TSDloss, we noticed that when the predicted bounding box is separated from the real bounding box, 1-TSD is used as the loss function. At this time, the loss function only restricts the center position of the two, and the size of the bounding box is not considered. Therefore, when two bounding boxes intersect or contain, the radius metric is added to constrain them. Due to TSDloss is a piecewise function. We want to know that in some boundary cases, such as when just intersected, TSDloss will the loss value mutate? If mutation occurs, will it have a negative impact on the results?

In addition, we think that the article still needs some supplementary experiments. In the current target detection field, some Loss functions have been designed for small target detection, such as Normalized Wasserstein Distance (NWD), Dot Distance (DotD), etc. Line 100 in this paper indicates whether "Observing that IoU and NWD does not did well in pest detection" is supported by research. We want to know the comparison between Loss suitable for small target detection and TSDloss proposed in this paper.

And we also noticed some textual errors:

1. In Figure 1., check whether the formula of TSD is correct.

2. Picture reference problem. For example, figure 7 is given on page 9, but is referenced for the first time on page 11. Figure 8 is given on page 10, but it is referenced for the first time on page 11.

3. Please check whether the reference method of "Tab 2" in row 262, "table 3" in row 270 and "Table 4" in row 278 meets the requirements.

Author Response

Dear reviewer,

Thank you for giving us the opportunity to submit a revised draft of the manuscript "TSD-Truncated Structural Aware Distance for Small Pest Object Detection". We appreciate the time and effort that you dedicated to providing feedback on our manuscript and are grateful for the insightful comments and valuable improvements to our paper. We have incorporated most of the suggestions made by the reviewer. Those changes are highlighted within the manuscript. Please see below, for a point-by-point response to your comments and concerns. All page numbers refer to the revised manuscript file with tracked changes.

Reviewer 3 Report

The research presents a new detecting algorithm for small objects such as pests. The proposed algorithm depends on considering the location of the object with respect to the ground trough to enhance its accuracy. The proposed algorithm includes not only assigning labels but also the NMS and the loss function. The proposed algorithm was tested using previously collected database and its result were compared with earlier detecting algorithms. The comparison indicated a slightly enhancing in the detecting accuracy.

The research is well organized, but the English style should be passive voice, please update all “We did…” and replace “our model or ours” with “the proposed model”. Moreover, the following comments must be considered before acceptance:

 Line 28,29 - use lowercase for “as Automatic Driving, Defect Detection, Aerial Image Analysis and Pest Detection.”

 Line 55 - add abbreviation to “SSD”

 Line 65 - add abbreviation to “GAN”

 Line 70 - add abbreviation to “AR-GAN”

 Line 85 - add abbreviation to “NMS”

 Fig. 2 – “A” group should be replaced by “Bgt” for TDS parts

 Fig. 3,5,7 – are unreadable

 What is “d” in Eq. 5?

 Line 172 – correct the spiling “tow” to “two”

 Eq. 6 is not looking mathematically right, how can “x” have 3 different formulas? Please explain.

 Line 201 – correct “five” to “four” as stages 2,3,4,5 are only four

 The descried parts of methodology in lines 219 to 222 are not included in Fig. 5

 The caption of Fig. 6 must contain the used method CE loss or Focal Loss or  …etc

 Line 269 – please explain the different categories of Pest24 database and why only 28 of them are considered in Table 3

 Fig.8 – correct the caption “second column” to “second row”, also “our” to “proposed”

 Line 276 – add abbreviations to IoU, GIoU, CIoU and DIoU

 It will be much better to replace Table 3 with graphical bar chart

 Table 4 – since there is no combinations between used methods, please replace the first 5 columns “with correct marks” with one column “with the considered methods vertically aligned”

Author Response

(The authors gave the same response as above.)

Round 2

Reviewer 2 Report

Thank you for your patient reply.

I still have some questions about TSDloss. When two bounding boxes are separated from each other, the size of TSDloss depends not only on the difference of bounding boxes, but also on the size of S value. Therefore, when the value of S is large, there are cases where the value of the formula is less than 1. Perhaps the probability of such a situation is small, but it should still be considered. And, when two bounding boxes intersect, as you mentioned "when they are just intersected, the value of cosθ is close to −1 and the value of formula is close to 2". So the loss value is discontinuous when the bounding boxes are in the boundary case of separation and intersection. Maybe this does not affect the convergence of the model, but in order to make the proposed loss function more reasonable, improvements can be considered in subsequent work.

I am very pleased to note that you have added NWD contrast experiment to table 4. It will be more convincing if the training log of NWD model is added to figure.8.

Author Response

Dear reviewer,

We appreciate the time and effort that you dedicated to providing feedback on our manuscript and are grateful for the insightful comments and valuable improvements to our paper. Please see below, for a point-by-point response to your further comments and concerns.

Comment 1

I still have some questions about TSDloss. When two bounding boxes are separated from each other, the size of TSDloss depends not only on the difference of bounding boxes, but also on the size of S value. Therefore, when the value of S is large, there are cases where the value of the formula is less than 1. Perhaps the probability of such a situation is small, but it should still be considered. And, when two bounding boxes intersect, as you mentioned "when they are just intersected, the value of cosθ is close to −1 and the value of formula is close to 2". So the loss value is discontinuous when the bounding boxes are in the boundary case of separation and intersection. Maybe this does not affect the convergence of the model, but in order to make the proposed loss function more reasonable, improvements can be considered in subsequent work.

Response:

We have also noticed the problem of discontinuity of loss value, but so far we have not found a way to improve it, and we will do further thinking and improvement in the follow-up research process.

Comment 2

I am very pleased to note that you have added NWD contrast experiment to table 4. It will be more convincing if the training log of NWD model is added to figure.8.

Response

We have added the AP and Loss of NWD model in Figure 8 (Page 10).

Thank you again for your approval of our work and suggestions.

Sincerely,

Xiaowen Huang

xwhuang@mail.ustc.edu.cn

Reviewer 3 Report

The authors carried out all the comments, however the text in the figures are still too small (unreadable). Accordingly the research is accepted after updating the figures.      

Author Response

Dear reviewer,

We appreciate the time and effort that you dedicated to providing feedback on our manuscript and are grateful for the insightful comments and valuable improvements to our paper. Please see below, for a point-by-point response to your further comments and concerns.

Comment

The authors carried out all the comments, however the text in the figures are still too small (unreadable). Accordingly the research is accepted after updating the figures.

Response:

We have adjusted the size of the text in the Figure 3, 5, 8 ( Page 4, 7, 10).

Thank you again for your approval of our work and suggestions.

Sincerely,

Xiaowen Huang

xwhuang@mail.ustc.edu.cn